# Stability and Antiproliferative Activity of Malvidin-Based Non-Oxonium Derivative (Oxovitisin A) Compared with Precursor Anthocyanins and Pyranoanthocyanins

**DOI:** 10.3390/molecules27155030

**Published:** 2022-08-07

**Authors:** Muci Wu, Yan Ma, Ao Li, Jingyi Wang, Jingren He, Rui Zhang

**Affiliations:** 1National R&D Center for Se-Rich Agricultural Products Processing, School of Modern Industry for Selenium Science and Engineering, Wuhan Polytechnic University, Wuhan 430048, China; 2Hubei Key Laboratory for Processing and Transformation of Agricultural Products, Wuhan Polytechnic University, Wuhan 430048, China; 3Hubei Key Laboratory of Natural Products Research and Development, College of Biological and Pharmaceutical Sciences, China Three Gorges University, Yichang 443002, China; 4School of Food and Biological Engineering, Hubei University of Technology, Wuhan 430068, China

**Keywords:** oxovitisin A, pyranoanthocyanin, stability, antiproliferative

## Abstract

Oxovitisins are a unique group of anthocyanin derivatives with a non-oxonium nature and α-pyranone (lactone) D ring on the structure. In this study, oxovitisin A was synthesized through the micro-oxidative reaction of carboxypyranomalvidin-3-O-glucoside (vitisin A) with water, and its thermostability, pH, and SO_2_ color stability were studied compared with its two precursors, malvidin-3-O-glucoside (Mv3glc) and vitisin A, as well as methylpyrano-malvidin-3-O-glucoside (Me-py). Results showed that oxovitisin A exhibited the highest stabilities, which were inseparably related to its noncharged structure and the additional carbonyl group on the D ring. Moreover, the antiproliferative capacity of oxovitisin A was comparatively evaluated against two human gastrointestinal cancer cell lines. Interestingly, oxovitisin A presented the strongest antiproliferative ability on MKN-28 (IC_50_ = 538.42 ± 50.06 μM) and Caco-2 cells (IC_50_ = 434.85 ± 11.87 μM) compared with two other pyranoanthocyanins. Therefore, we conclude that oxovitisin A as a highly stable anthocyanin derivative still exhibits a satisfactory antiproliferative ability.

## 1. Introduction

Anthocyanins are the most important polyphenolic natural pigments in the plant kingdom and are highly valued for their functional activities, such as antioxidant [1], anticancer [2], antiatherosclerosis [3], and prevention of diabetes [4]. However, native anthocyanins are unstable and susceptible to degradation by pH variations, light, temperature, solvents, and chemical structure, limiting their industrial applicability [4,5].

Pyranoanthocyanins are an important group of anthocyanin-derived nature pigments that have an additional pyranic D ring in the anthocyanin structure; they were first identified in red wine and had greater stability than native anthocyanins [6,7]. Among these derivatives, different substituents on the C-10 position of a new pyranic ring may lead to differences in stability. Schwarz and Winterhalter reported that pinotin A was less susceptible to pH shifts and retained its original color over a wide pH variance [8]. He et al. found that pyranoanthocyanin-flavanols had superior pH stability and entirely SO_2_ bleaching resistance compared with malvidin-3-O-glucoside (Mv3glc) and carboxypyranomalvidin-3-O-glucoside (vitisin A). After a 6-month storage, pyranoanthocyanin-flavanols were proved to be more stable against degradation than native anthocyanin in the following order: pyranoMv3glc-catechin > pyranoMv3glc-dimer B3 > pyranoMv3glc-epicatechin > vitisin A > Mv3glc [9]. Recently, Sun et al. investigated that vitisin A with a carboxy group on the D ring had the greatest pH and SO_2_ color stability compared with Mv3glc and other five pyranoanthocyanins, and methylpyrano-malvidin-3-O-glucoside (Me-py) had the greatest thermostability compared with other pigments involved [10]. However, although oxovitisins are a group of pyranoanthocyanin-derived compounds with a unique non-oxonium nature and additional pyranone D ring (Figure 1), there are only a few publications reporting the identification and formation of these neutral anthocyanin derivatives [11,12], as well as their antioxidant efficiency measured through radical scavenging experiments and theoretical approaches [13,14]. Relevant literature concerning their stability is absent.

Anthocyanins have been reported to act in the inhibition of key modulators that promote cancer progression and development [2,15,16]. Publications have demonstrated that anthocyanins have a selective antiproliferation effect on tumor cells while having relatively little or no effect on the growth of normal cells [2]. It has been reported that anthocyanins have an effect on preventing gastric cancer cell proliferation [17,18]. In the stomach, an important digestive organ, 20–25% of anthocyanins are absorbed into the blood circulation in a complete structure within a few minutes after ingestion through a gastric site [19]. The structural influence on their transport efficiency has also been explored by a gastric cell model. Previous research investigated that oxovitisin A had the highest transport efficiency through the gastric cancer cell monolayers in comparison with pyranoanthocyanins and native anthocyanin [20]. Therefore, the antiproliferative effect of oxovitisin A on gastric tumor cells would be interesting with high gastric transport efficiency. After gastric absorption, the remaining anthocyanins are rapidly absorbed and quickly metabolized in the intestine; both intact forms and metabolites, including methylated, glucuronidated, or sulfated derivatives, are appear in the circulation [19]. The effects of anthocyanins on the prevention of intestinal cancer have been widely assayed [21,22,23], and anthocyanins can inhibit the proliferation of Caco-2 cells by promoting intracellular reactive oxygen species, inducing caspase-3 activation and upregulating the expression of a cyclin-dependent kinase inhibitor [22]. Nevertheless, publications regarding the biological features of pyranonthocyanins with greater stability are lacking, especially oxovitisin A with a similar structure to flavones.

Due to the important role and potentiality of anthocyanins and pyranoanthocyanins as natural pigment dyes and functional active substances, it is crucial to advance the knowledge of the stability and biological activity of derivatives with special structural features. However, related studies focusing on the systematic comparison of a non-oxonium derivative with its precursors are lacking. Therefore, the stabilities of oxovitisin A based on Mv3glc were evaluated in this study, including pH, SO_2_, and thermal stability on color characteristics, and compared with another common pyranoanthocyanin (Me-py) and precursor anthocyanins, including vitisin A and Mv3glc. In addition, the antiproliferative effects of oxovitisin A and respective pigments on gastric and intestinal cancer cell lines were comparatively researched. The aim of this work was to elucidate the effects of the non-oxonium nature and different substituents at the C-10 position on the stability and antiproliferative activity of anthocyanin and derivatives, and is further expected to provide a theoretical basis for the application of non-oxonium anthocyanin derivatives in food and health products.

## 2. Results and Discussion

### 2.1. Stability of Oxovitisin A

The spectral properties of oxovitisin A, vitisin A, Me-py, and Mv3glc were characterized in relation to temperature, pH, and bisulfite decoloration. In all the assays, oxovitisin A had the highest stability compared with its precursors and Me-py.

### 2.2. Thermostability

The thermal stability of oxovitisin A and other pigments was explored from 50 to 90 °C. Oxovitisin A was more stable than two pyranoanthocyanins and Mv3glc at the same temperature, as presented in Appendix A. After heating at 50 °C for 1 and 5 h, the color value of oxovitisin A at λ_max_ only fell to 99.61% and 99.48%, respectively. Moreover, the reservation rate of oxovitisin A can be maintained at 91.35% after 5 h of heating at 90 °C; Me-py and vitisin A still had relatively high residual rates at 86.30% and 72.6%, respectively, whereas the value of Mv3glc dropped to 29.52%.

The line relationship between the logarithm of pigment contents −ln(*C_t_*/*C_0_*) and heating time *t* followed the first-order reaction kinetic (Figure 2). At lower heating temperatures, three pyranoanthocyanins can maintain high stability except for Mv3glc. Oxovitisin A exhibited a higher stability than other pyranoanthocyanins and monomeric anthocyanin at higher temperatures. Calculated kinetic parameters are given in Appendix A. The values of the first-order reaction kinetic constants *k* rose with increasing temperature, whereas half-lives *t_1/2_* decreased. It is indicated that higher temperatures can accelerate the degradation of anthocyanins and induce lower stability. This can be explained by the fact that the chemical structure damage of anthocyanins would increase with the temperature and heating time rising, and eventually degrade to colorless chalcone and coumaric glycosides or aldehyde and benzoic acid derivatives [24]. The results correspond to a recent publication on the thermal stability of pyranoanthocyanins based on Mv3glc [10]. Among the four pigments tested, oxovitisin A showed the greatest stability against thermal treatment with significantly longer *t_1/2_* values, followed by pyranoanthocyanins, including Me-py and vitisin A; Mv3glc exhibited the worst thermal stability. The results indicate that the newly formed D ring on the structure can significantly improve the stability of pyranoanthocyanins. Furthermore, the different groups linked on the C-10 position of the D ring could impact the thermal stability (carbonyl group > methyl group > carboxyl group). In addition, the neutral structure of oxovitisin A may have a positive contribution to its thermal stability.

### 2.3. pH Stability

The UV–visible absorption spectrum of four pigments with different pH values is shown in Figure 3. The λ_max_ of Mv3glc, vitisin A, Me-py, and oxovitisin A were 518, 498, 470, and 373 nm, respectively, at acidic pH 2.5. In comparison with precursors, the λ_max_ of oxovitisin had an obviously blue shift at the UV region, the absorbance of oxovitisin A showed a steady increase, and the λ_max_ and color intensity were nearly no variance with the increase in pH values (pH 1.0–7.0) from acid to neutral conditions. This indicated that oxovitisin A was extremely stable in the test pH range. The λ_max_ showed a stable bathochromic shift at higher pH values, including pH 9.0 (384 nm) and pH 11.0 (414 nm), with the absorbance increased significantly at pH 11.0 and the waveform kept smooth, and oxovitisin A displayed an unusual and interesting amber yellow color (Appendix A). In contrast, the λ_max_ of Mv3glc, vitisin A, and Me-py were detected in visible range and bathochromic shift during the increase in pH from 1.0 to 11.0. In acid solution (pH 1.0–2.5), Mv3glc showed a red color, and its color was nearly light and disappeared above pH 3.6 and significantly turned to blue above 9.0 (Appendix A). These results correspond with those of Sun and Oliveira, which were mentioned previously [6,10]. Similarly, the absorbance of Mv3glc dropped quickly as the pH increased to pH 3.6 at a colorless form and finally rose after neutral pH conditions. Vitisin A was reported to have a better pH resistance than precursors Mv3glc for its newly formed new pyranic ring D [10]. In our assay, vitisin A showed a better color resistance than Mv3glc; it can maintain λ_max_ around 504 nm below pH 7.0; after that, the absorbance decreased steadily, and the pigment solvents eventually turned to light blue above pH 9.0 with λ_max_ bathochromic shift. The color change discipline of Me-py was similar to that of vitisin A; this pigment exhibited a stable yellow color below pH 9.0, and turned to purple at higher pH values. Under acidic to neutral solvent conditions, the color changes from red to light and then to blue with the absorbance decreasing constantly, indicating that the precursor anthocyanins form a colorless hemiacetal structure under the nucleophilic attack of the solvent, and further transfer to a blue quinoidal base form. In this case, oxovitisin A showed better pH stability than its two precursors and Me-py.

All the pigment solutions above were submitted to *L*a*b** color measurements; the color properties of oxovitisin A and its precursors are summarized in Table 1 based on *L** (correlate of lightness), *C** (correlate of saturation), and *h_a,b_* (hue angle). *L** represents lightness to darkness (+*L* to *−L*), *+a* represents redness to greenness −*a*, +*b* represents yellowness to blueness −*b*, +*C* represents neutral chroma to fully saturated color −*C*, and *h* represents the hue angle of color change around the color circle (+*h* or −*h*) [6].

According to the obtained results, oxovitisin A presented similar values of lightness (*L**) and was higher than Mv3glc, vitisin A, and Me-py in pH ranges, indicating that oxovitisin A can maintain high and stable color lightness. *L** values were slightly decreased, meaning that oxovitisin A solution was slightly darker at pH 9.0 and 11.0. The chroma values (*C**) of oxovitisin A were basically unchanged from pH 1.0 to 4.5 and slightly increased in neutral pH values (5.5 and 7.0) and, finally, rose significantly after pH 9.0. This reveals that the saturation of oxovitisin A solutions was stable in acidic condition and slightly increased in neutral condition; then the solutions were significantly saturated in an alkaline environment and exhibited a rich intensity of amber yellow color. Contrastingly, other three pigments had the highest chroma value at acidic solutions and decreased at high pH values. Compared with Mv3glc, Me-py and vitisin A still have a high saturated color at pH 3.6–7.0, which is attributed to the pyranoid ring that stabilizes the pigment molecule by preventing the hydration of pyranoanthocyanins to a colorless carbinol base or hemiacetal form. However, the chroma values of these two pyranoanthocyanins dropped significantly at higher pH values (pH > 7.0) because the positively charged structure was more susceptible to nucleophilic attack than oxovitisin A. All the oxovitisin A solutions were detected to have the negative *a* value and a greenness property compared with two precursors. On the other hand, all the oxovitisin A solutions showed higher positive *b* values, which revealed a yellow-dominated color. The +*b* values increased constantly at pH 1.0–7.0 and obviously rose after pH 9.0; the solvent colors were observed from light yellow to saturated dark yellow. Oxovitisin A presented stable positive values of the hue angle (*h_a,b_*). When compared with higher pH values, the *h_a,b_* values were decreased, which meant that the color of oxovitisin A shifted toward the blue pole of the chromatic circle. Similarly, vitisin A solutions at pH 3.6–5.5 presented higher values of *h_a,b_* compared with pH 2.5, which means that the color shifted toward the yellow pole of the chromatic circle; thus, it could exhibit an orange-red-like color. For Me-py, *h_a,b_* values at acidic pH conditions (1.0–3.6) were nearly unchangeable, but decreased with increasing pH values, thereby shifting the color of the solution to the blue pole of the chromatic circle. Δ*E* means the total chromatic difference of test samples towards the reference sample (pH 1.0). The colorimetric difference Δ*E** reveals the changes in all colorimetric parameters, such as chroma, lightness, and tonality [6]. According to this result, the color of oxovitisin A does not have a significant difference at pH 1.0 to 7.0, indicating that it has a much better color stability in acidic and neutral solutions than the two precursors and Me-py. Overall, oxovitisin A showed the greatest color stability at a wide pH range (1.0–11.0); it only exhibited yellow color and gloss, increasing within higher pH values. The chemical structure is the main contributor to this property. In comparison with the other three pigments, oxovitisin A does not have the positive charge in its structure that could make it better to avoid nucleophilic attacks from small molecules. Moreover, the *a*-pyranone (lactone) ring between C-4 and the hydroxy group at C-5 of the anthocyanin core induces a high degree of conjugation and delocalization of *π* electrons, which increases structural stability to prevent the influence of pH medium [13].

### 2.4. Bisulfite Bleaching Stability

The bleaching effect of SO_2_ on the color stability of oxovitisin A was studied, and all the anthocyanin derivatives showed greater resistance to SO_2_ when compared with Mv3glc in the range of 0–200 ppm (Figure 4). The results of vitisin A, Me-py, and Mv3glc corresponded to previous publications [6,9,10]. The color intensity and absorbance of anthocyanin derivatives decreased slightly, whereas the Mv3glc solution became almost uncolored at a higher concentration of SO_2_ (200 ppm), and the absorbance decreased significantly. These results are in agreement with the respective bleaching constants (*K_SO2_*) calculated. Indeed, the *K_SO2_* obtained for oxovitisin A (0.061 × 10^−3^) was significantly lower than vitisin A (0.14 × 10^−3^), Me-py (0.632 × 10^−3^), and Mv3glc (3.592 × 10^−3^). Oxovitisin A exhibited the strongest ability against SO_2_, indicating its stable chemical structure. The reason is that with C-2 and C-4 being the two main chromogenic positions in the anthocyanin structure, the nucleophilic addition of anionic bisulfite occurs at these positively charged positions and induces the pigment decolorization of anthocyanins [6]. Concerning oxovitisin A, vitisin A, and Me-py, the newly formed pyran ring on the C-4 position could prevent the nucleophilic attack of bisulfite in that position. The relatively small decrease in the color intensity observed for vitisin A and Me-py can be explained by the nucleophilic attack of bisulfite at the less favored positive C-2 position [6]. In this assay, no visible color decolorization was observed for oxovitisin A, which can be attributed to its unique flavone-like structure with higher stability. Moreover, the non-oxonium-natured structure without positive charge could be better in avoiding nucleophilic attack from small molecules [12].

### 2.5. Antiproliferative Capacity

The antiproliferative capacity of oxovitisin A, Mv3glc, vitisin A, and Me-py was evaluated by the sulforhodamine B (SRB) method. The antiproliferative results were expressed as IC_50_ values and are shown in Figure 5. After continuous exposure for 48 h, all compounds exhibited a dose-dependent growth inhibitory effect. Mv3glc and oxovitisin A exhibited the strongest inhibitory effect on MKN-28 cancer cell proliferation with IC_50_ values of 332.65 ± 11.62 μM and 538.42 ± 50.06 μM, respectively. No significant difference was observed between vitisin A and Me-py, but significant differences were found between oxovitisin A and these two pyranoanthocyanins (*p* < 0.01). The antiproliferative effects of compounds on MKN-28 cells were as follows: Mv3glc > oxovitisin A > Me-py > vitisin A.

In the stomach, an important digestive organ of animals, a high proportion of anthocyanin glycosides (approximately 25%) or red orange anthocyanins (about 20%) can be quickly and efficiently absorbed via the stomach with intact structure; these bioactive anthocyanins could be the first act on stomach tissue after digestion [19]. Moreover, Fernandes et al. suggested that the antiproliferative properties of anthocyanins were related to their structures. Conjugation with methyl groups alters the number of hydroxyl and methoxyl groups in ring B, which can decrease the antiproliferative properties of original anthocyanins on human cancer cells [18]. Sousa et al. compared the antiproliferative properties of different desoxyanthocyanidins and anthocyanins, and all the compounds tested showed antiproliferative activity on stomach (AGS, MKN-28) and intestinal (Caco-2) cancer cell lines, especially against Caco-2 cell growth [25]. Our previous work also assessed the influence of structure on the antiproliferative ability between malvidin-based anthocyanins and derivatives and discovered that Mv3glc and oxovitisin A had a greater effect in inhibiting the proliferation of human breast cancer cells (MCF-7) than vitisin A and Me-py [20]. In this assay, the antiproliferative effect of pigments on MKN-28 cells is inseparable from their structure. Mv3glc does not have the additional D ring, so it has an extra −OH group compared with other compounds; publications have reported that the number and position of hydroxyl groups can directly influence the antioxidant and bioavailability [5,16,26]. On the other hand, Mv3glc as the original anthocyanin has the smallest structure size, making it easier to diffuse and incorporate into the cancer cell lines to express antiproliferative ability. Among the derivatives, oxovitisin A is the most effective compound in inhibiting the proliferation of MKN-28 cells. These results suggest that the different substituents on the D ring may have a direct impact on the antiproliferative ability against MKN-28 cancer cells. Oxovitisin A, which has a carbonyl group on the pyran ring, is the most effective antiproliferative derivative, followed by vitisin A with a carboxyl group and Me-py with a methyl group as the least effective. Furthermore, previous research has proved that oxovitisin A has the highest transport efficiency among these derivatives to cross the gastric barrier [20]. This may be attributed to its special structure without the positive charge, which can lead to a more stable compound once it becomes hindered from nucleophilic attacks, and the additional pyranone ring could make the conjugation structure more stable.

The antiproliferative capacity of pigments on Caco-2 showed similar results as follows: Mv3glc > oxovitisin A > vitisin A > Me-py (Figure 5). Mv3glc and oxovitisin A were the most effective inhibitors, with IC_50_ values of 309.88 ± 20.17 μM and 434.85 ± 11.87 μM, respectively. Sousa et al. suggested that the lack of glucose and hydroxy groups can clearly decrease the antiproliferative activity of anthocyanins on cancer cells, and all the anthocyanins were active against Caco-2 cells other than gastric cancer cell lines (AGS and MKN-28) [25]. These results are in accordance with our study, since Mv3glc with an extra hydroxyl moiety exhibits the greatest antiproliferative effect against MKN-28 and Caco-2 cell lines compared with other derivatives. Furthermore, all the compounds showed a higher antiproliferative activity against Caco-2 than MKN-28, and the IC_50_ values in our assay were in the range of published data [17,25].

Overall, anthocyanins have been reported to have various bioactivities, but their bioavailability is considered to be low, especially at the intestinal level (<1%) [27,28], limiting their beneficial effects as a consequence of rapid absorption and elimination in the body [19], some critical factors that may contribute to this paradox, such as the effect of pH, temperature, the ability of compounds to cross membranes, and digestive enzymes [4,19,29]. Their structures are unstable, especially in the weakly alkaline environment of the small intestine (pH 7.4), and carbinol (anionic) pseudo base and quinoidal forms are the main structures present [30]. Furthermore, anthocyanins are poorly absorbed as genuine parent glycosides or detected in the blood as metabolites [29,31], despite recent publications indicating that anthocyanin metabolites with a significant extent could also display considerable biological function activities at several degrees and targets [4,18,19,31]. In comparison with monomeric anthocyanins, oxovitisin A as a highly stable derivative, and it can maintain the original chemical structure and exhibit an excellent color feature even in alkaline environments and higher temperatures, thus leading to no changes in bioactivities. This study also demonstrated that oxovitisin A has a considerable antiproliferative ability on MKN-28 and Caco-2 cells. Besides, previous publication also proved that this special anthocyanin derivative has excellent transport efficiency and could be rapidly absorbed into breast cancer cells after only 4 min [20]. Otherwise, the IC_50_ values of each compound on MCF-7 cell line inhibition are significantly lower than the effect on gastrointestinal cancer cell lines, suggesting that even an obviously lower concentration of pigments could reach the breast tissue due to digestion and bloodstream journey losses, but they can still perform anticancer abilities in vivo at a low content [20].

## 3. Materials and Methods

### 3.1. Materials

TSK Toyopearl gel HW-40(s) was purchased from Tosoh (Tokyo, Japan); pyruvic acid, sulforhodamine B, acetone, and ethanol of analytical grade were purchased from Sigma-Aldrich (Madrid, Spain). All solvents of HPLC grade were provided by Thermo Fisher Scientific (Waltham, MA, USA). The cancer cell lines MKN-28 and Caco-2 were purchased from the China Center for Type Culture Collection (Wuhan, China).

### 3.2. Synthesis of Vitisin A, Me-Py, and Oxovitisin A

Grape skin extract was obtained from Yunnan Tonghai Yang Natural Products Co., Ltd. Tonghai, China, and Mv3glc was the main anthocyanin component as previously detected [32]. An Mv3glc fraction was preliminarily extracted by a method described by Oliveira et al. [20]; grape skin extract was dissolved in acidic aqueous solution and applied directly onto a polyamide gel column (mesh 80–100, 30 mm × 600 mm I.D.). Briefly, a 5% (*v*/*v*) methanol acidic aqueous solution was first applied to remove the impurities, the fractions of original anthocyanins containing Mv3glc were eluted with an increasing 10% methanol acidic aqueous solution. After evaporation under vacuum to remove the solvents, the collected fractions were freeze-dried and stored at −20 °C until further purification.

The preparation of vitisin A was performed by using a method described by Oliveira et al. [6], and a Mv3glc fraction and pyruvic acid were mixed with a molar ratio of 50:1 in water (pH 2.6, 35 °C) for 5 days. The resulting fraction was preliminarily purified by polyamide resin (10% acid ethanol solution for impurity removal, 20% acid ethanol solution for elution) for the next reaction.

Me-py was formed by the reaction of the Mv3glc fraction and acetone with a ratio of 15:1 (*w*/*v*), as previously reported [32], and the mixed aqueous solution was kept at 45 °C and pH 3.0 for 9 days, after which the reaction was stopped and the major fraction containing Me-py was purified.

Oxovitisin A was achieved via a slow oxidation of vitisin A (1 mg/mL) in aqueous ethanol solution, with 20% ethanol at pH 3.6 45 °C for 21 days according to the reported procedure with modification [12,20]. All the synthesized products were monitored, and their chemical structures were characterized by HPLC/DAD-MS method reported before [33]; data are shown in Appendix A. The chemical structures of Mv3glc, vitisin A, Me-py, and oxovitisin A are given in Figure 1.

### 3.3. Purification of Mv3glc and Its Derivatives

The further purification of Mv3glc was carried out using an SSI Series 1500 semi-preparative HPLC equipped with a 250 × 10 mm I.D. 5 μm reversed-phase C18 column (Allsphere ODS-2) and a UV detector at 35 °C. The chromatographic elution program was used as previously described [33], with minor modification. The eluent solvents were A, H_2_O/HCOOH (9:1), and B, H_2_O/CH_3_CN/HCOOH (6:3:1). The gradient consisted of 15%30% of B for 30 min and then to 100% after 31 min. The sample injected volume was 5 mL and eluted at a rate of 4 mL/min. The fractions eluted were detected at 540 nm.

Following preliminary purification by polyamide resin column, vitisin A fraction was vacuum-concentrated and submitted to the semipreparative HPLC described above, with the eluent solvents and gradient modified as follows: A, H_2_O/HCOOH (9:1), and B, H_2_O/CH_3_CN/HCOOH (1:8:1). The solvent consisted of 15–21% B during 30 min and increased to 100% B at 32 min. The separated vitisin A was detected at 510 nm.

The purification of Me-py was followed by the method of Zhu et al. [32]. First, the mixture solution was directly loaded on the polyamide resin column, and Me-py fraction was eluted with 10% acid ethanol solution. The fraction was further loaded onto a TSK Toyopearl gel HW-40(S) column (40 mm × 600 mm I.D.) and eluted with 10–20% acid methanol solution. The yellow fraction was purified by the semipreparative HPLC with the modification of eluent solvents and gradient: A, H_2_O/HCOOH (9:1), and B, H_2_O/CH_3_CN/HCOOH (1:8:1). The solvent consisted of 20–30% B during 30 min and increased to 100% B at 33 min. The separated Me-py was detected at 478 nm.

The synthetic oxovitisin A was first submitted to the same resin column above. After total absorption, the column was washed with 30% (*v*/*v*) methanol acidic aqueous solution to remove nonreacted pigments, and the remaining fraction was eluted by 40% (*v*/*v*) methanol acidic aqueous solution. The collected fraction was purified by a TSK Toyopearl gel HW-40(S) column (40 mm × 600 mm I.D.) with 40% (*v*/*v*) methanol acidic aqueous solution to remove the residual vitisin A; then a light-yellow fraction of oxovitisin was eluted by 50% (*v*/*v*) methanol acidic aqueous solution. Finally, separated oxovitisin A was obtained using semipreparative HPLC under the same conditions as vitisin A above, except that the eluent gradient consisted of 18–35% B over 50 min, increasing to 100% B at 52 min. The isolated yellow fraction was collected under 373 nm.

All the synthesized pigments were concentrated and lyophilized. The final purities of pigments involved are all above 98%, which were confirmed by HPLC–DAD analysis, as measured by the ratio of the target peak area to the total peak at respective λmax [24]. After purification, these powders were stored at −20 °C until use.

### 3.4. Thermal Stability

Oxovitisin A, Me-py, vitisin A, and Mv3glc were dissolved in 50 mL of citro-phosphate buffer solutions (0.1 mM, pH 2.2). An amount of 10 mL of each sample was heated in a water bath for 5 h at different temperatures from 50 to 90 °C. The absorption value of each sample was measured every hour after immediately cooling to room temperature.

The first-order reaction kinetic rate constants (*k*) and half-lives (*t_1/2_*) were calculated using the following Equations (1) and (2):(1)$$-\ln\left(C_{t}/C_{0}\right)=k\times t$$
(2)$$t_{1/2}=-\ln0.5\times k^{-1}$$
where *C_0_* is the initial anthocyanin content, and *C_t_* is the anthocyanin content after *t* hours of heating at a given temperature [34,35].

### 3.5. pH and Color Stability

Mv3glc and its derivatives including oxovitisin A were chromatically characterized. Aqueous solutions of each pigment (0.1 mM) were prepared at different pH values between 1.0 and 11.0 adjusted by HCl or NaOH. All the prepared solutions were balanced under natural light for 15 min, and then the color was observed, and the spectral absorbance curves were also recorded from 300 to 700 nm using a 10 × 10 mm cell in an Evolution 220 UV spectrophotometer (Thermo, Waltham, MA, USA). Moreover, the color properties of solutions with different pH values were explored by an UltraScan VIS Spectrophotometer (HunterLab, Reston, VA, USA), and the CIELAB coordinates (*L**, *a** and *b**) were measured and automatically calculated [6].

### 3.6. SO_2_ Bleaching Stability

The effect of SO_2_ on the color stability of oxovitisin A and other pigments was also explored by using the pigment solutions at pH 2.2 (0.1 mM), and different aliquots of an aqueous solution of sodium bisulfite (5 mg/mL) were added into pigment solutions to obtain SO_2_ concentrations ranging from 0 to 200 ppm [9]. The spectroscopic absorbance curves of all the mixed solutions were recorded using a 10 mm × 10 mm cell in an Evolution 220 UV spectrophotometer with a 1 nm sample interval after they were balanced under natural light for 15 min. The bleaching constant (*K_SO2_*) was obtained from the slope of color intensity (at the λ_max_ of each pigment) as a function of the concentration of SO_2_.

### 3.7. Cell Culture Conditions

Two human cancer cell lines were cultured according to a previously published method [17]. MKN-28 and Caco-2 cells were grown as monolayers in 25 cm^2^ plates and maintained in DMEM and MEME and supplemented with 10% and 15% FBS, respectively, 1% antibiotic/antimycotic solution (100 units per mL of penicillin; 100 μg/mL of streptomycin; and 0.25 μg/mL of amphotericin B) at 37 °C under a humidified atmosphere with 5% CO_2_. Cancer cells were harvested by trypsinization (0.25% (*w*/*v*) trypsin–EDTA_4_Na) twice a week.

### 3.8. Sulforhodamine B Assay

The antiproliferative effects of pigments on cancer cells (MKN-28 and Caco-2 cell lines) were evaluated by using the protein-binding dye SRB to assess cell growth, according to the procedure described by Oliveira et al. [17,20]. Briefly, cells (1.5 × 10^5^ cells/mL) were spread into 96-well plates and allowed to grow for 24 h before treatment in order to ensure exponential growing state. The cancer cells continued to be cultured and exposed to a range concentrations of oxovitisin A for 48 h (50–750 μM for Caco-2 and 70–1125 μM for MKN-28), and each compound concertation was replicated for 6 holes. This value was chosen based on the previously reported value that appropriately 0.4 mg/mL of anthocyanins (around 750 μM) can reach the stomach after the intake of a glass of red wine (200 mL) [17,18]. Thus, the concentration of pigments was selected comprehensively in consideration of their dilution in a gastric content.

The incubation was terminated with the addition of TCA 50% for 1 h at 4 °C, and the plates were washed by distilled water containing 1% of acetic acid. The plates were then dried and stained by 0.4% of SRB solution for 30 min. After that, the dye was removed and the plates were washed with acidic water and dried again. At last, the remaining dye was eluted with Trizma buffer (10 mM, pH 10.5) for 30 min in the dark. The absorbance of each hole was measured by a multimode microplate reader (EnSpire, PerkinElmer, MA, USA) at 492 nm. Mv3glc, vitisin A, and Me-py were all tested at the same concentration for comparison.

### 3.9. Statistical Analysis

The experiment results are expressed as the means ± standard error mean (SEM). The statistical significance of the difference between groups was determined by one-way analysis of variance (ANOVA), followed by the Bonferroni post-test. Differences were considered to be statistically significant when * *p* < 0.05 vs. oxovitisin A. All the statistical analysis was performed using GraphPad Prism 9.0.0 (GraphPad Software, Inc., La Jolla, CA, USA).

## 4. Conclusions

In this study, we systematically explored the stabilities and antiproliferative effects of the non-oxonium derivative oxovitisin A in comparison with precursor anthocyanins and Me-py. The interesting outcome is that oxovitisin A is extremely stable under physical factors, including temperature, pH, and SO_2_, compared with native anthocyanins and other proanthocyanins, which means that this compound can maintain its color and structure in harsh environments during manufacturing, shipping, and storage processes and further exhibit bioactivities. Moreover, oxovitisin A being the most stable anthocyanin, can keep a considerable antiproliferative ability compared with precursor anthocyanins, indicating the excellent application prospects of this compound in food, cosmetic, and health products.

## Figures and Tables

**Figure 1 molecules-27-05030-f001:**
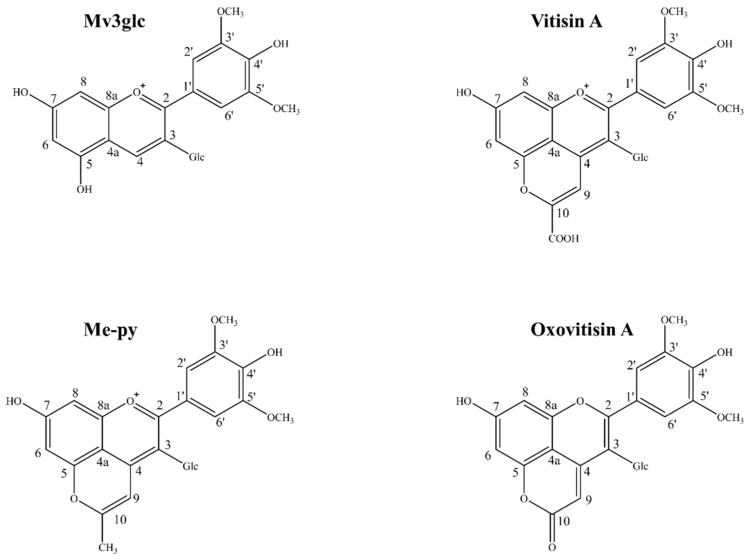
Chemical structures of Mv3glc, vitisin A, Me-py, and oxovitisin A.

**Figure 2 molecules-27-05030-f002:**
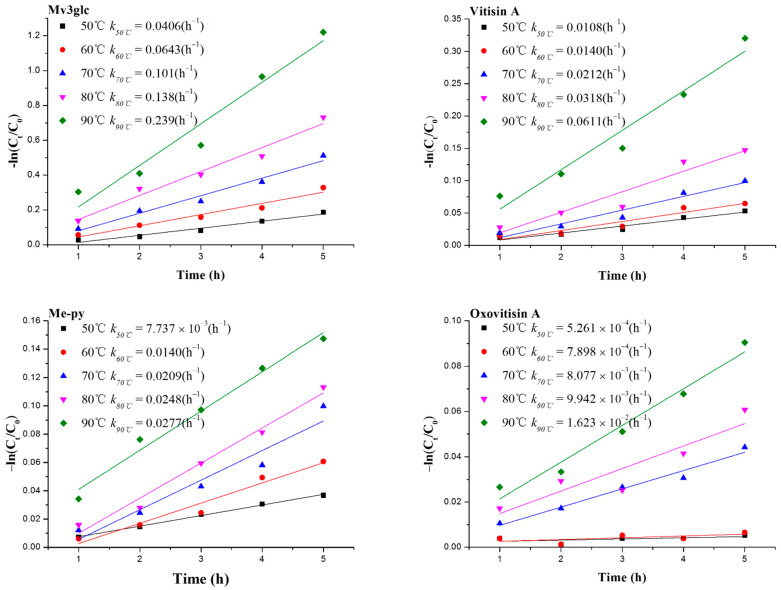
Degradation kinetic analysis of Mv3glc, vitisin A, Me-py, and oxovitisin A under different temperatures (50, 60, 70, 80, 90 °C).

**Figure 3 molecules-27-05030-f003:**
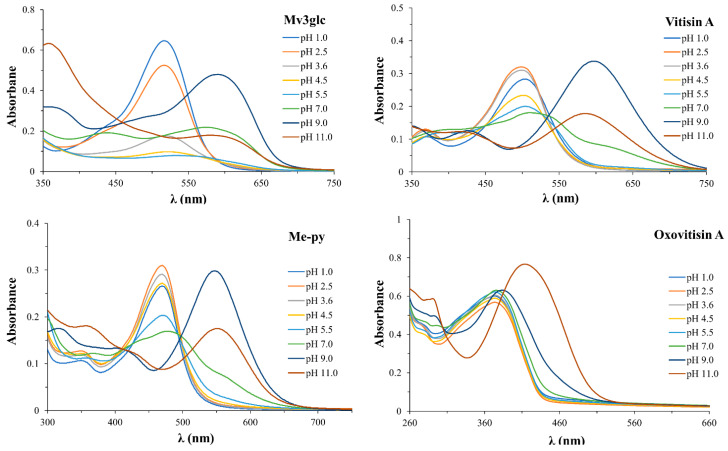
UV–VIS spectra of Mv3glc, vitisin A, Me-py, and oxovitisin A at different pH values (1.0–11.0).

**Figure 4 molecules-27-05030-f004:**
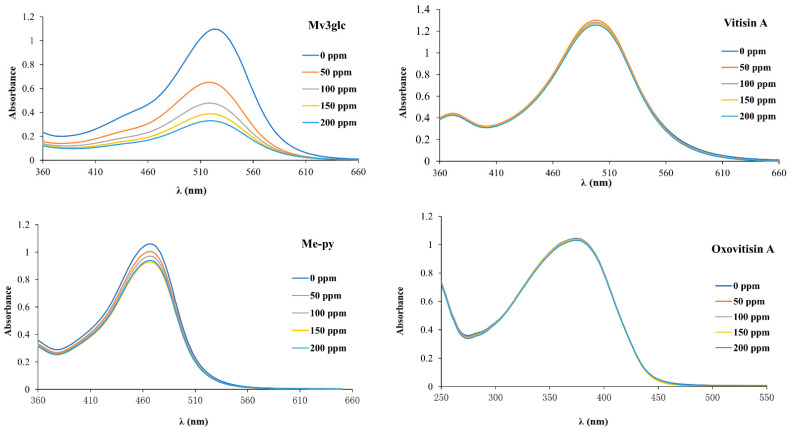
Visible spectral of Mv3glc, vitisin A, Me-py, and oxovitisin A under different concentrations of bisulfite.

**Figure 5 molecules-27-05030-f005:**
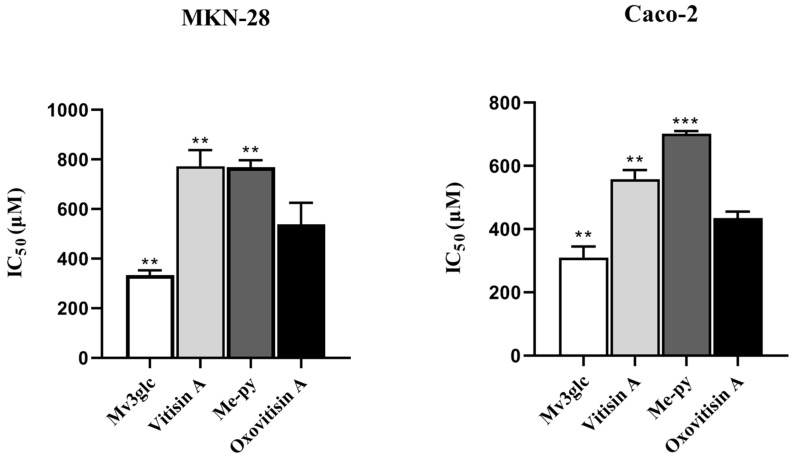
Antiproliferative abilities of Mv3glc, vitisin A, Me-py, and oxovitisin A against MKN-28 and Caco-2 cell lines in vitro. Each value represents the means ± SEM (*n* = 6). ** *p* < 0.01, *** *p* < 0.001 vs. oxovitisin A.

**Table 1 molecules-27-05030-t001:** Color coordinates of Mv3glc, vitisin A, Me-py, and oxovitisin A at different pH values.

	**Mv3glc**								
	** *L** **	** *a** **	** *b** **	** *C** **	** *h_a,b_* **	** Δ*L** **	** Δ*C** **	** Δ*h_a,b_* **	** Δ*E** **
pH 1.0	29.12	7.54	−0.05	7.54	359.61	-	-	-	-
pH 2.5	30.21	5.45	−0.43	5.47	355.46	1.08	−2.07	−4.15	2.39
pH 3.6	31.22	2.30	−0.55	2.36	346.62	2.09	−5.18	−12.99	5.67
pH 4.5	31.19	1.39	−0.73	1.57	332.3	2.07	−5.97	−27.31	6.53
pH 5.5	31.28	1.21	−0.76	1.43	327.95	2.16	−6.11	−31.65	6.73
pH 7.0	31.35	1.44	−0.69	1.60	334.39	2.22	−5.94	−25.21	6.53
pH 9.0	31.37	0.51	−0.52	0.73	314.70	2.25	−6.81	−44.91	7.40
pH 11.0	29.03	−1.64	1.11	1.98	145.91	−0.1	−5.56	−213.7	9.26
	**Vitisin A**								
	** *L** **	** *a** **	** *b** **	** *C** **	** *h_a,b_* **	** Δ*L** **	** Δ*C** **	** Δ*h_a,b_* **	** Δ*E** **
pH 1.0	23.74	7.52	−2.58	7.95	341.06	-	-	-	-
pH 2.5	24.76	8.96	0.7	8.99	4.47	1.02	1.04	−336.59	3.72
pH 3.6	25.52	8.84	2.85	9.28	17.87	1.78	1.33	−323.19	5.86
pH 4.5	25.71	9.16	3.1	9.67	18.71	1.97	1.72	−322.35	6.23
pH 5.5	25.25	7.63	1.92	7.87	14.13	1.51	−0.08	−0.08	4.75
pH 7.0	24.11	3.49	0.03	3.49	0.54	0.37	−4.46	−4.46	4.82
pH 9.0	23.65	1.31	−0.68	1.48	332.57	−0.09	−6.47	−6.47	6.49
pH 11.0	23.45	−2.92	−1.51	3.29	207.37	−0.29	−4.66	−4.66	10.50
	**Me-py**								
	** *L** **	** *a** **	** *b** **	** *C** **	** *h_a,b_* **	** Δ*L** **	** Δ*C** **	** Δ*h_a,b_* **	** Δ*E** **
pH 1.0	31.85	0.89	11.28	11.31	85.51	-	-	-	-
pH 2.5	31.83	0.76	11.32	11.35	86.15	−0.02	0.04	0.64	0.13
pH 3.6	31.8	0.89	11.2	11.23	85.45	−0.05	−0.08	−0.06	0.1
pH 4.5	31.41	1.9	10.21	10.38	79.47	−0.44	−0.93	−6.04	1.54
pH 5.5	31.17	2.52	9.43	9.76	75.01	−0.68	−1.55	−10.5	2.57
pH 7.0	30.76	3.16	8.24	8.83	69.05	−1.09	−2.84	−16.46	3.94
pH 9.0	27.16	4.35	1.10	4.48	14.27	−4.69	−6.83	−71.24	11.72
pH 11.0	25.18	4.53	−4.64	6.49	314.33	−6.67	−4.82	228.82	17.64
	**Oxovitisin A**								
	** *L** **	** *a** **	** *b** **	** *C** **	** *h_a,b_* **	** Δ*L** **	** Δ*C** **	** Δ*h_a,b_* **	** Δ*E** **
pH 1.0	33.99	−1.31	2.07	2.45	122.38	-	-	-	-
pH 2.5	34.01	−1.42	2.27	2.68	122.07	0.02	0.23	−0.31	0.23
pH 3.6	34.06	−1.47	2.27	2.7	123.01	0.07	0.25	0.63	0.27
pH 4.5	34.1	−1.51	2.26	2.71	123.73	0.11	0.26	1.35	0.29
pH 5.5	34.04	−1.89	3.54	4.02	118.12	0.05	1.57	−4.26	1.58
pH 7.0	33.93	−1.85	3.32	3.8	119.13	−0.06	1.35	−3.25	1.37
pH 9.0	33.33	−2.78	9.82	10.2	105.8	−0.66	7.75	−16.58	7.92
pH 11.0	32.71	−1.74	12.81	12.93	97.74	−1.28	10.48	−24.64	10.83

## Data Availability

Data are contained within the article.

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
