# Peer review of "Stability and Antiproliferative Activity of Malvidin-Based Non-Oxonium Derivative (Oxovitisin A) Compared with Precursor Anthocyanins and Pyranoanthocyanins"

_molecules, 2022, doi:10.3390/molecules27155030_

Round 1
Reviewer 1 Report
The manuscript of Wu et al. deals with the stability and antiproliferative activity of malvidin-based non-oxonium derivative (Oxovitisin A) compared to precursor anthocyanins and pyranoanthocyanins.
It is important that the authors fully justify why it is important to perform these stability tests compared to what should be reported for similar compounds.
For each test, the rationale for the results needs to be discussed from a broader chemical point of view.
An illustrative scheme of the syntheses described in 3.2 is desirable.
The conclusions sound like a repetition of the results. The authors should explain what the reader gains from knowing that they did all these tests and how this adds to their knowledge of anthocyanins.
In my opinion, the manuscript in general looks like a routine assay that can be carried out in a week, unless the authors fully justify why these measurements deserve to be published in Molecules.
Reviewer 2 Report
The presented manuscript describes the stability and anti-proliferative activity of oxovitisin in comparison to its precursors anthrocyanins (vitisin A and methylpyrannomalvidin-o-gycoside) and malvidin3-o-glycoside.
The coloured pigments are abundant in the plant kingdom and several researchers have investigated their stabilities, bioavailability, and their functional activities. The focus of the presented manuscript was to study the pH, thermal stability and bisulfide bleaching stability of oxovitisin in comparison to the other derivatives to guide towards its use in cosmetic and dietary products.
The methods presented for stability testing were sufficient to compare the 4 described derivatives and to prove the superiority of oxovitisin compared to its precursors and Malvidin-3-o-glycoside. Pigments colour stability study at different pH and the bisulfide bleaching stability testing provided sufficient evidence to support the findings of improved stability of oxovisitin.
The antiproliferative activity was not as strong in this study. The authors investigated the effect of all 4 compounds on two cancer cell lines, MKN-28 (gastric cancer cell line) ad Caco-2 (intestinal cancer cell line). The IC50 reported were quite high (more than 500 µM) which doesn’t qualify oxovitisin to be regarded as having good antiproliferative activity. The antiproliferative activity of these compounds against these cancer cell lines have been previously investigated and didn’t add much to the study.
An improved approach towards investigating the effect of these compounds in cancer would be carrying out an in vivo study that proves their ability to reach these regions to induce their antiproliferative effect.
There is also no justification of the improved antiproliferative activity of oxovitisin which could be due to improved permeability or higher activity. Comparing the mechanism of action and the permeability of these compounds could provide better insight into the lower IC50 values for oxovitisin.
I recommend the authors add extra investigations of the permeability of Oxovitisin into the studied cancer cell lines and compare these with the other three compounds.
The improved thermal, pH and bisulfide bleaching stability of oxovitisin A would be very beneficial if accompanied with evidence of its superior permeability into cancer cells.
Reviewer 3 Report
In the manuscript submitted to me for review entitled: "Stability and antiproliferative activity of malvidin based non-oxonium derivative (oxovitisin A) compared to precursor anthocyanins and pyranoanthocyanins", oxovitisin A was synthesized and some of its characteristics were investigated, in which it performed well in comparison with its two precursors malvidin-3-O-glucoside (Mv3glc) and vitisin A, as well as methylpyrano-malvidin-3-O-glucoside (Me-pee). The antiproliferative activity of oxovitisin A against two human gastrointestinal colon cancer cell lines - MKN-28 and Caco-2 cells - was also determined, which turned out to be the most significant compared to the other two investigated pyranoanthocyanins.
My questions and remarks to the authors are:
1. The manuscript abstract is not written according to the editors' requirements. It should represent a passage containing the subsections: 1) Background; 2) Methods; 3) Results and 4) Conclusion. View instructions for authors.
2. It is not clear what the "*" symbol in Table 1 represents. Usually, these symbols are given below the table.
3. On line 228, the abbreviation "SRB method" is entered. This is the first time this abbreviation appears in the manuscript and must be accompanied by the full method name.
4. How do the MKN-28 and Caco-2 cell lines selected for antiproliferative activity differ? Do they give significant differences in the study of antiproliferative activity in other studies of the authors. My question is because both cell lines are from human gastro intestinal cancer. It is not stated in materials and methods where the two cell lines were obtained from.
Round 2
Reviewer 1 Report
The manuscript is now suitable for publication in Molecules.